# Diagnostic Accuracy of Dehydroepiandrosterone Sulfate and Corticotropin in Autonomous Cortisol Secretion

**DOI:** 10.3390/biomedicines9070741

**Published:** 2021-06-28

**Authors:** Lindsay E. Carafone, Catherine D. Zhang, Dingfeng Li, Natalia Lazik, Oksana Hamidi, Maria Daniela Hurtado, William F. Young, Melinda A. Thomas, Benzon M. Dy, Melanie L. Lyden, Trenton R. Foster, Travis J. McKenzie, Irina Bancos

**Affiliations:** 1Department of Internal Medicine, May Clinic, Rochester, MN 55902, USA; carafone.lindsay@mayo.edu; 2Division of Endocrinology, Mayo Clinic, Rochester, MN 55902, USA; zhang.catherine@mayo.edu (C.D.Z.); li.dingfeng@mayo.edu (D.L.); lazik.natalia@mayo.edu (N.L.); Oksana.Hamidi@UTSouthwestern.edu (O.H.); Hurtado.Mariadaniela@mayo.edu (M.D.H.); young.william@mayo.edu (W.F.Y.J.); thomas.melinda@mayo.edu (M.A.T.); 3Division of Endocrinology and Metabolism, UT Southwestern Medical Center, Dallas, TX 75390, USA; 4Division of Endocrinology, Mayo Clinic Health System, La Crosse, WI 54601, USA; 5Department of Surgery, Mayo Clinic, Rochester, MN 55902, USA; dy.benzon@mayo.edu (B.M.D.); Lyden.melanie@mayo.edu (M.L.L.); Foster.Trenton@mayo.edu (T.R.F.); mckenzie.travis@mayo.edu (T.J.M.)

**Keywords:** adrenal adenoma, adrenal mass, dexamethasone suppression test

## Abstract

Autonomous cortisol secretion (ACS) affects up to 50% of patients with adrenal adenomas. Despite the limited evidence, clinical guidelines recommend measurement of serum concentrations of dehydroepiandrosterone-sulfate (DHEA-S) and corticotropin (ACTH) to aid in the diagnosis of ACS. Our objective was to determine the accuracy of serum concentrations of DHEA-S and ACTH in diagnosing ACS. We conducted a retrospective single center study of adults with adrenal adenoma evaluated between 2000−2020. Main outcome measure was diagnostic accuracy of DHEA-S and ACTH. ACS was defined as post-dexamethasone cortisol >1.8 mcg/dL. Of 468 patients, ACS was diagnosed in 256 (55%) patients with a median post-DST cortisol of 3.45 mcg/dL (range, 1.9–32.7). Patients with ACS demonstrated lower serum concentrations of DHEA-S (35 vs. 87.3 mcg/dL, *p* < 0.0001) and ACTH (8.3 vs. 16 pg/mL, *p* < 0.0001) compared to patients with non-functioning adrenal tumors (NFAT). Serum DHEA-S concentration <40 mcg/dL diagnosed ACS with 84% specificity and 81% PPV, while serum ACTH concentration <10 pg/mL diagnosed ACS with 75% specificity and 78% PPV. The combination of serum concentrations of DHEA-S <40 mcg/dL and ACTH <10 pg/mL diagnosed ACS with the highest accuracy with 92% specificity and 87% PPV. Serum concentrations of DHEA-S and ACTH provide additional value in diagnosing ACS.

## 1. Introduction

Adrenal tumors are common incidental findings, detected on 7% of cross-sectional abdominal imaging studies [1]. The majority are benign adrenal cortical adenomas without overt hormone excess; however, up to 30–50% of patients with incidentally discovered adrenal tumors (AI) have mild autonomous cortisol secretion [1,2,3,4], which is characterized by cortisol excess from adrenal adenomas or hyperplasia without the obvious physical manifestations of Cushing syndrome (e.g., facial plethora, proximal muscle weakness, supraclavicular fat pads, and wide purple-red striae) [5]. Recent studies have increasingly linked mild autonomous cortisol secretion to a variety of adverse clinical outcomes including abnormal body composition, increased risk of obesity, dyslipidemia, type 2 diabetes mellitus, osteoporosis, frailty, hypertension, cardiovascular events, and overall mortality [6,7,8,9,10]. Current clinical practice guidelines recommend that all patients with adrenal tumors undergo an overnight 1 mg dexamethasone suppression test (DST) to assess for cortisol excess, where post-DST serum cortisol concentration ≤1.8, 1.9–5.0, and >5.0 mcg/dL correspond to excluded, possible (not definite), and confirmed autonomous cortisol secretion (ACS), respectively [2,3]. For patients with post-DST serum cortisol concentrations in the range of 1.9–5.0 mcg/dL, in attempt to confirm and further characterize the degree of ACS, guidelines recommend additional measurement for corticotropin (ACTH) and dehydroepiandrosterone sulfate (DHEA-S), as well as repeating the DST [2]. While current guidelines recommend the use of these biomarkers to confirm the diagnosis, evidence on the diagnostic performance of serum concentrations of DHEA-S and ACTH for ACS is very limited.

Several small and heterogeneous studies have assessed the accuracy of serum DHEA-S concentration in diagnosing ACS, with discrepant results. One study demonstrated a high diagnostic performance of age and sex adjusted DHEA-S ratio [11], however accuracies of serum DHEA-S concentrations in diagnosing ACS were lower in three other studies [12,13,14]. Similar to serum DHEA-S concentration, the diagnostic performance of serum ACTH concentration in ACS is not well established. In addition to small sample sizes in existent studies, confident conclusions and comparisons are not possible due to variability in definition of ACS, laboratory assays for serum DHEA-S and ACTH concentrations, and the cutoffs applied.

Our primary objective was to determine the accuracy of serum concentrations of DHEA-S and ACTH in diagnosing ACS.

## 2. Materials and Methods

This is a retrospective study of adult patients with adrenal adenoma or hyperplasia evaluated at our institution, between 1 January 2000 and 20 April 2020. The study was approved by the Institutional Review Board, and only patients who provided authorization for use of their medical records were included. Medical records were reviewed for clinical, biochemical, and imaging data.

Patients were included if they had all three of the following: (1) unilateral or bilateral adrenal adenoma(s) or hyperplasia identified by computed tomography or magnetic resonance imaging, (2) measurement of serum cortisol after 1-mg DST, and (3) measurement of serum DHEA-S at the time of initial evaluation. Patients were excluded if they had any of the following: (1) exogenous glucocorticoid use within 3 months prior to testing, (2) adrenal insufficiency of any type, (3) ACTH-dependent Cushing syndrome, (4) overt Cushing syndrome, (5) adrenal mass other than adrenal adenoma, micronodular or macronodular hyperplasia, (6) congenital adrenal hyperplasia, (7) polycystic ovarian syndrome, or (8) features of androgen excess (Figure 1).

Subgroup analysis was performed in patients with available ACTH measurements.

Using the reference standard of post-DST cut-off of serum cortisol concentration >1.8 mcg/dL, patients with post-DST serum cortisol concentrations >1.8 mcg/dL and lacking overt features of cortisol excess were classified as having ACS (disease positive = ACS). Patients with post-DST serum cortisol concentrations ≤1.8 mcg/dL were classified as having nonfunctioning adrenal tumors (NFAT).

### 2.1. Biochemical Analysis

DHEA-S was measured by the chemiluminescent competitive binding immunoenzymatic assay (Siemens Immulite until 28 May 2018, followed by the Access DHEA-S, Beckman-Coulter Inc., Fullerton, CA, USA 2017 until the end of study). The inter-assay precision for the Siemens Immulite assay was <9% coefficient of variation and for the Beckman Coulter assay was <7% coefficient of variation, and comparison between the two assays on 50 samples was excellent (R^2^ of 0.98). In the interpretation of results, DHEA-S was analyzed as both absolute values and DHEA-S ratio (DHEA-S/DHEA-S lower limit of sex and age reference range), based on the reference values reported in Appendix A.

Plasma serum ACTH concentrations were measured by the electrochemiluminescence immunoassay (Siemens Immulite until 9th August 2017 with the reference range of 10–60 pm/mL, and Elecsys ACTH, Roche Diagnostics, Indianapolis, IN 2017 until the end of study, with the reference range 7.2–63 pg/mL for morning serum ACTH concentration. The inter-assay precision for the Siemens Immulite assay was <5% coefficient of variation and for the Roche assay was <3% coefficient of variation, and comparison between the two assays on 54 samples was excellent (R^2^ of 0.91). Cortisol was measured by the competitive binding immunoenzymatic assay (Access Cortisol, Beckman-Coulter, Brea, CA, USA 2007), with coefficient of variation of <8%.

### 2.2. Statistics

Statistical analysis was performed using JMP Software, Version 15 (SAS, Carey, NC, USA). Categorical data were reported as absolute and relative frequencies, and continuous data were presented as medians with ranges. Subgroup analyses were performed using a *t*-test for continuous variables and using a Chi-Squared test for categorical variables. Linear regression was performed to determine associations between continuous variables. Statistical significance was determined by *p* value less than 0.05. Receiver operating characteristic (ROC) curves were generated with determination of area under the curve (AUC) and confidence intervals.

## 3. Results

### 3.1. Patients

Of the 468 patients included in the analysis, 308 (66%) were women, and the median age of diagnosis of adrenal adenoma was 58 years (range 18–89). Adenomas were unilateral in 73% of patients (left 44%). The median size was 26 mm (range, 5–135), Table 1.

There were 256 (55%) patients diagnosed with ACS and 212 (45%) with NFAT. Median post-DST serum cortisol concentration was 3.5 mcg/dL (range, 1.9–32.7) in patients with ACS compared to 1.2 mcg/dL (range, 0.5–1.8) in patients with NFAT.

Patients with ACS were slightly older than patients with NFAT (median age of 59 vs. 57 years, *p* = 0.0005), and proportion of women was similar (68% vs. 63%, *p* = 0.2). Patients with ACS had a much higher prevalence of bilateral disease (34% vs. 19% in NFAT, *p* = 0.0005) and demonstrated a larger tumor size (median of 34 mm vs. 19 mm, *p* < 0.0001), Table 1.

Adrenalectomy was performed in 160 patients with ACS–unilateral in 150, and bilateral in 10 patients, Table 1.

### 3.2. Accuracy of DHEA-S and ACTH in Diagnosing of ACSs

Serum DHEA-S concentration (35 vs. 87 mcg/dL, *p* < 0.0001) and DHEA-S ratio (2.3 vs. 4.6, *p* < 0.0001) were significantly lower in patients with ACS vs. NFAT. Serum ACTH was measured in 371 (79%) patients (154 patients with NFAT and 217 patients with ACS), and was lower in patients with ACS when compared to patients with NFAT (median of 8.3 vs. 16 pg/mL, *p* < 0.0001), Table 1.

The proportion of low serum DHEA-S and ACTH concentrations was higher in patients with ACS (Figure 2A,B). Post-DST serum cortisol concentration was inversely associated with both serum DHEA-S and ACTH concentrations (Figure 2C,D), and serum DHEA-S concentrations correlated with serum ACTH concentrations (Figure 2E).

ROC curves were generated for multiple DHEA-S cutoffs to determine the best threshold for diagnosing ACS, Table 2. The cutoff which performed best was serum DHEA-S concentration <40 mcg/dL, with a specificity of 83.5% and positive predictive value (PPV) of 80.8%. DHEA-S ratios did not demonstrate superior diagnostic performance.

We analyzed serum ACTH concentration cutoffs of <10, <15 and <20 pg/mL for diagnosing ACS, Table 2. The cutoff which performed best in diagnosing ACS was serum ACTH concentration <10 pg/mL, with a specificity of 75.3% and PPV of 77.9%.

The combination of serum concentrations of DHEA-S and ACTH for diagnosing ACS was analyzed at various cutoffs, Table 2. Serum DHEA-S concentration <40 mcg/dL and serum ACTH concentration <10 pg/mL diagnosed ACS with 91.6% specificity, 86.6% PPV, and 8.4% false positive rate. Diagnostic performance was similar in patients with mild ACS (post-DST cortisol of 1.9–5 mcg/dL), Appendix A. On the other hand, the combination of serum DHEA-S concentration >100 mcg/dL and ACTH >15 pg/mL demonstrated a specificity of 96% and PPV of 80.4% in excluding ACS, with a false positive rate of 4.1%, Appendix A.

## 4. Discussion

In this study, we found that serum DHEA-S concentration cutoff of <40 mcg/dL and serum ACTH concentration cutoff of < 10 pg/mL diagnose ACS with similar accuracy. Accuracy is improved when measurements of DHEA-S and ACTH are used in combination.

We found that a serum concentration of DHEA-S < 40 mcg/dL diagnoses ACS with 83.5% specificity and 80.8% PPV. Sex- and age-adjusted analysis revealed similar results. It is difficult to compare our results to previous studies due to different definitions of ACS (Table 3).

For example, a study of 38 patients with ACS defined by post-DST serum cortisol concentration of >3 mcg/dL and either low ACTH or elevated 24-h urine free cortisol demonstrated that serum DHEA-S concentration cutoff of 40 mcg/dL performed with 75% specificity and 43% PPV [13]. In another study of 41 patients with ACS (defined as post-DST serum cortisol concentration >1.8 mcg/dL) and 11 patients with NFAT, serum DHEA-S concentration cutoff of 66 mcg/dL demonstrated a sensitivity of 93% and specificity of 82%, but comparisons to our cohort are limited by a very small sample size as well as differences in the assay used to measure DHEA-S [12]. In a study of 58 patients with ACS (defined as post-DST serum cortisol concentration >1.8 mcg/dL) and 54 patients with NFAT, serum DHEA-S concentration cutoff of 40 mcg/dL demonstrated a sensitivity of 58% and specificity of 80%, similar to our results [14]. In the absence of an adrenal disorder, biosynthesis of DHEA-S by the adrenal cortex declines with age, and men have higher serum DHEA-S levels than women. In a cohort of 167 patients with an adrenal mass (29 patients with ACS), a DHEA-S ratio ≤ 1.12 demonstrated >99% sensitivity and 92% specificity in diagnosing ACS [11]. Notably, in addition to a small sample size, all included patients with ACS in this study had serum ACTH concentration < 10 mcg/dL, and the definition of ACS was based on presence of at least two abnormalities of hypothalamic-pituitary-adrenal (HPA) axis assessment [11]. In our study, as well as in another smaller study [14], DHEA-S ratios did not perform better compared to the serum DHEA-S concentration cutoff of 40 mcg/dL, suggesting that the decrease in serum DHEA-S in ACS is more dependent on the degree of adrenal cortex atrophy independent of sex and age. DHEA-S ratios are cumbersome to use in clinical practice and require the clinician to take additional steps to calculate the ratio correctly based on the appropriate reference value. This also carries risk of human error in calculations. Thus, using the absolute DHEA-S value would simplify interpretation of results and avoid misclassification without decrease in accuracy.

We found that a serum ACTH concentration cutoff of <10 pg/mL diagnoses ACS with 75% specificity and 78% PPV. Our results were similar to a prior study which demonstrated that “low” ACTH had a good diagnostic accuracy for ACS; however, the absolute cutoff for ACTH was not defined in that study [15]. Diagnosis of ACS was most accurate in our study when combining serum concentrations of DHEA-S and ACTH. The combination of DHEA-S < 40 mcg/dL and ACTH < 10 pg/mL diagnosed ACS with 91.6% specificity and 86.6% PPV, which was improved from the performance of these biomarkers when used individually. This combination also achieved a low false positive rate of 8.4%. This was improved from the false positive rates of 16.5% with serum DHEA-S concentration < 40 mcg/dL and 24.7% with serum ACTH concentration < 10 pg/mL when used alone.

The combination of elevated serum concentrations of DHEA-S and ACTH excluded ACS with high accuracy, with the best performing cutoffs of DHEA-S > 100 mcg/dL and ACTH > 15 pg/mL. Thus, in this setting, proceeding with additional testing to diagnose ACS may be unnecessary.

### 4.1. Strengths and Limitations

Our study is the first large scale study to investigate the diagnostic accuracy of serum concentrations of DHEA-S and ACTH in diagnosing ACS as defined by the recent guidelines. Patients were recruited from a single institution, which allowed for uniformity of biomarker assays so that differences in results would not be potentially attributable to variations in assays. Our study had several limitations. Our cohort comprised patients who had DHEAS measurements during initial evaluation, which could have led to selection bias. This was a retrospective study which resulted in some missing variables for ACTH. We have relied on the accuracy of post-DST serum cortisol concentration when classifying patients into ACS, however false positive DST results can occur, resulting in a possible misclassification of some patients with ACS. The cutoffs for serum concentrations of ACTH and DHEA-S we propose may not be applicable to centers that use different assays.

### 4.2. Clinical Implications

Our study suggests that in patients with post-DST serum cortisol concentration > 1.8 mcg/dL, obtaining baseline serum ACTH and DHEA-S (on a separate day than DST) is valuable as this can avoid repeating DST, or performing additional tests for cortisol excess. This diagnostic approach may also be beneficial in patients with suspected false positive results due to use of oral contraceptives, rapid metabolism, non-compliance, or other issues. Furthermore, we propose that when baseline hormonal evaluation reveals serum DHEA-S concentration >100 mcg/dL and serum ACTH concentration >15 pg/mL, clinicians could avoid the inconvenience of completing the overnight 1-mg DST to diagnose ACS.

## 5. Conclusions

In conclusion, serum concentrations of DHEA-S and ACTH separately, and in combination, provide additional value when diagnosing ACS.

## Figures and Tables

**Figure 1 biomedicines-09-00741-f001:**
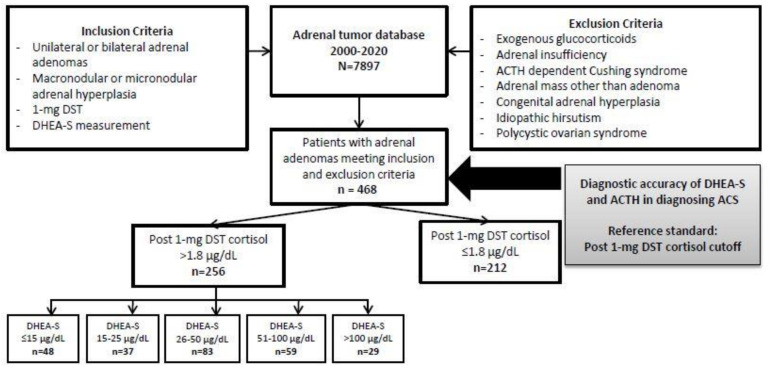
Flowchart of study population. Abbreviations: ACTH, corticotropin; DHEA-S, dehydroepiandrosterone sulfate; DST, dexamethasone suppression test; ACS, mild autonomous cortisol secretion; NFAT, nonfunctioning adrenal tumor.

**Figure 2 biomedicines-09-00741-f002:**
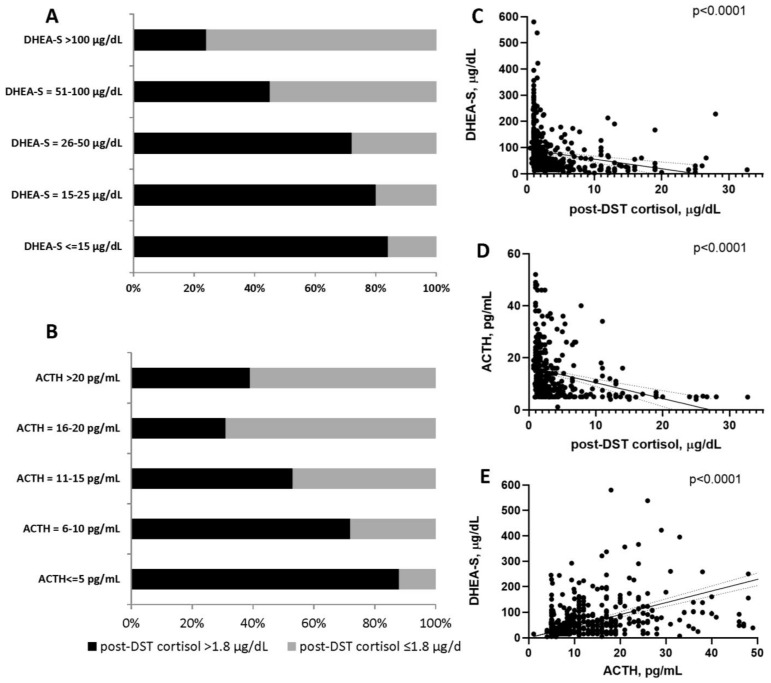
(**A**) Comparison of serum DHEA-S concentrations to 1-mg overnight dexamethasone suppression test (DST) results for post-DST serum cortisol concentrations of ≤1.8 versus >1.8 mcg/dL; (**B**) Comparison of serum ACTH concentrations to 1-mg overnight dexamethasone suppression test (DST) results for post-DST serum cortisol concentrations of ≤1.8 versus >1.8 mcg/dL. of what is contained in the second panel. (**C**) Association between serum DHEA-S concentration and post-DST serum cortisol concentration. (**D**) Association between serum ACTH concentration and post-DST serum cortisol concentration. (**E**) Association between serum DHEA-S and ACTH concentrations. Abbreviations: ACTH, corticotropin; DHEA-S, dehydroepiandrosterone sulfate; DST, dexamethasone suppression test; ACS, mild autonomous cortisol secretion; NFAT, nonfunctioning adrenal tumor.

**Table 1 biomedicines-09-00741-t001:** Clinical and biochemical presentation of patients with adrenal adenomas based on the 1-mg overnight dexamethasone suppression test (DST).

	Total	Post-DST Cortisol ≤1.8 mcg/dL	Post-DST Cortisol >1.8 mcg/dL	*p* Value
	(*n* = 468)	(*n* = 212)	(*n* = 256)	
Age, years				
Median (ranges)	58 (18–89)	57 (20–89)	59 (18–87)	0.0005
Female, *n* (%)	308 (66)	133 (63)	175 (68)	0.2
Bilateral, *n* (%)	126 (27)	40 (19)	86 (34)	0.0005
Tumor size, mm				
Median (ranges)	26 (5–135)	19 (5–135)	34 (5–128)	<0.0001
Post-DST cortisol, mcg/dL				
Median (ranges)	2.0 (0.5–32.7)	1.2 (0.5–1.8)	3.45 (1.9–32.7)	
DHEA-S, mcg/dL				
Median (ranges)	53 (2.8–606)	87.3 (11–606)	35 (2.8–244)	<0.0001
DHEA-S ratio ^a^	3.2 (0.1–33.6)	4.6 (0.6–33.6)	2.3 (0.1–23.3)	<0.0001
Median (ranges)				
ACTH ^b^, pg/mL	11 (1.1–52)	16 (4.9–52)	8.3 (1.1–46)	<0.0001
Median (ranges)				
Adrenalectomy, *n* (%)	219 (47)	59 (28)	160 (63)	<0.0001

^a^ DHEA-S ratio was calculated as (DHEA-S)/(DHEA-S lower limit of sex- and age- reference range). ^b^ Available in 371 patients (154 patients with post-DST cortisol ≤1.8 mcg/dL and 217 patients with post-DST cortisol >1.8 mcg/dL). ACTH, corticotropin; DHEA-S, dehydroepiandrosterone sulfate; DST, dexamethasone suppression test.

**Table 2 biomedicines-09-00741-t002:** Diagnostic accuracy parameters for serum DHEA-S and ACTH concentration cutoffs in diagnosing autonomous cortisol secretion.

Cutoff	Sensitivity, % (95% CI)	Specificity, % (95% CI)	PPV, % (95% CI)	NPV, % (95% CI)	False Positive Rate, %
DHEA-S					
<15 mcg/dL	18.8 (14.2–24.1)	95.8 (92.1–98.0)	84.2 (72.8–91.4)	49.4 (47.8–51.0)	4.2
<25 mcg/dL	33.2 (27.4–39.3)	91.5 (86.9–94.9)	82.5 (74.6–88.4)	53.2 (50.8–55.5)	8.4
<40 mcg/dL	57.4 (51.1–63.6)	83.5 (77.8–88.2)	80.8 (75.3–85.3)	61.9 (58.2–65.5)	16.5
<50 mcg/dL	65.6 (59.5–71.4)	76.4 (70.1–82.0)	77.1 (72.2–81.3)	64.8 (60.5–68.9)	23.6
<80 mcg/dL	83.2 (78.1–87.6)	53.8 (46.8–60.6)	68.5 (65.1–71.7)	72.6 (66.3–78.2)	46.2
<100 mcg/dL	88.7 (84.1–92.3)	42.9 (36.2–49.9)	65.2 (62.4–68.0)	75.8 (68.3–82.1)	57.0
DHEA-S ratio					
<1.2	32.0 (26.4–38.1)	92.9 (88.6–96.0)	84.5 (76.5–90.2)	53.1 (50.8–55.4)	7.0
<1.8	42.2 (36.1–48.5)	85.4 (79.9–89.8)	77.7 (70.9–83.3)	55.0 (52.1–57.9)	14.6
ACTH					
<10 pg/mL	61.8 (54.9–68.3)	75.3 (67.7–81.9)	77.9 (72.4–82.6)	58.3 (53.6–62.9)	24.7
<15 pg/mL	79.3 (73.3–84.5)	53.9 (45.7–62.0)	70.8 (66.8–74.4)	64.8 (57.8–71.3)	46.1
<20 pg/mL	87.6 (82.4–91.6)	27.3 (20.4–35.0)	62.9 (60.3–65.4)	60.9 (50.1–70.7)	72.7
DHEA-S and ACTH					
DHEA-S < 40 mcg/dL and ACTH < 10 pg/mL	38.7 (32.2–45.5)	91.6 (86.0–95.4)	86.6 (78.9–91.8)	51.5 (48.6–54.4)	8.4
DHEA-S < 40 mcg/dL and ACTH < 15 pg/mL	49.3 (42.5–56.2)	88.3 (82.2–92.9)	85.6 (79.1–90.4)	55.3 (51.7–58.8)	11.7
DHEA-S < 40 mcg/dL and ACTH < 20 pg/mL	53.9 (47.0–60.7)	85.7 (79.2–90.8)	84.2 (78.0–88.9)	56.9 (53.0–60.7)	14.3
DHEA-S < 25 mcg/dL and ACTH < 10 pg/mL	26.3 (20.5–32.7)	94.8 90.0-97.7	87.7 (77.8–93.6)	47.7 (45.5–49.9)	5.2
DHEA-S < 25 mcg/dL and ACTH < 15 pg/mL	31.3 (25.2–38.0)	93.5 (88.4–96.8)	87.2 (78.4–92.7)	49.2 (46.7–51.6)	6.5
DHEA-S < 25 mcg/dL and ACTH < 20 pg/mL	34.1 (27.8–40.8)	91.6 (86.0–95.4)	85.1 (76.6–90.8)	49.7 (47.0–52.3)	8.4

**Table 3 biomedicines-09-00741-t003:** Studies investigating accuracy of DHEA-S in diagnosing autonomous cortisol secretion (ACS) ^a^.

Author, Year	Patients ^a,b^ *n*	Diagnosis of ACS	DHEA-S Assay	DHEA-S Cutoff	Accuracy
Yener, 2015 [13]	38 ACS 141 NFAT	At least 2 of the following: Post-DST cortisol >3 mcg/dL ACTH <10 pg/mL 24 h urine cortisol >70 mcg/dL	Solid-phase, competitive, chemiluminescent enzyme immunoassay (Immulite 2000, Diagnostic Products Corporation, Los Angeles, CA, USA)	DHEA-S of 40 mcg/dL	AUC of 0.79 Sensitivity of 68% Specificity of 75%
Dennedy, 2017 [11]	29 ACS 138 NFAT	At least 2 of the following: Post-DST cortisol >1.8 mcg/dL Sleeping midnight cortisol >1.8 mcg/dL or awake midnight cortisol >7.5 mcg/dL Elevated 24 h urine cortisol All patients had ACTH <10 pg/mL (not a definition criteria)	Solid-phase competitive immunoassay, Siemens (Surrey, UK) Immulite 2000 platform	DHEA-S ratio ≤1.12	AUC of 0.95 Sensitivity of 100% Specificity of 91.9%
Ueland, 2020 [14]	58 ACS 54 NFAT	Post-DST cortisol >1.8 mcg/Dl [2]	Chemiluminescent immunoassay (CLIA) using Siemens Immulite 2000 XPi	DHEA-S of 40 mcg/dL	AUC of 0.76 Sensitivity of 58% Specificity of 80%
				DHEA-S ratio	AUC of 0.69
Current study	256 ACS 212 NFAT	Post-DST cortisol >1.8 mcg/dL [2]	Chemiluminescent competitive binding immunoenzymatic assay (Access DHEA-S, Beckman-Coulter Inc., Fullerton, CA, USA 2017)	DHEA-S of 40 mcg/dL DHEA-S ratio	AUC of 0.767 Sensitivity of 57.4% Specificity of 83.5% AUC of 0.693

^a^ Only studies with at least 20 patients with ACS published since 2015, DHEA-S measurement with immunoassay are included. ^b^ Only patients with both post-DST cortisol and DHEAS are included. Abbreviations: ACTH, corticotropin; DHEA-S, dehydroepiandrosterone sulfate; DST, dexamethasone suppression test; NFAT: nonfunctioning adrenal tumor; NPV: negative predictive value, PPV: positive predictive value.

## Data Availability

Data are available upon reasonable request to the corresponding author.

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
