# Peer review of "Diagnostic Accuracy of Dehydroepiandrosterone Sulfate and Corticotropin in Autonomous Cortisol Secretion"

_biomedicines, 2021, doi:10.3390/biomedicines9070741_

Round 1
Reviewer 1 Report
The study by Carafone and colleagues reports the results of a retrospective evaluation of 468 patients with benign adrenal tumors. The main aim is to evaluate the accuracy of serum DHEAs and ACTH in diagnosing autonomous cortisol secretion. The study is interesting and well written. Here my suggestions and comments.
- According to the primary objective of the study (accuracy of serum DHEAs and ACTH in diagnosing ACS), the analysis of the study should be restricted to the 371 patients with both ACTH and DHEAs measurements. Indeed, measurement of ACTH values should be listed in the inclusion criteria.
- How was ACTH-dependency excluded in the 39 patients with ACS without ACTH measurements?
- DHEAs is a key measurement in this paper. Please, provide more details about the assay (coefficient of variation?)
- Please, clarify whether the assays for DHEAs, ACTH and cortisol were unchanged during the last 20 years (time of the retrospective data collection).
Author Response
Thank you for you effort reviewing and providing valuable suggestions.
We have addressed all points and include a word document with our responses.

Reviewer 2 Report
The Authors address the interesting issue of the evaluation of diagnostic accuracy of ACTH and DHEA-S levels in the definition of autonomous cortisol secretion. They suggest that serum concentrations of DHEA-S and ACTH provide additional value in diagnosing ACS.
However, some concerns derive from reading the study.
- The Authors should specify if patients with overt CS were excluded. Please comment that there are some patients with very high cortisol levels after dex test and also some very large adrenal masses.
- Have lab methods never changed for 20 years?
- Absolute values of DHEA-S levels are not clinically useful, since they are age-dependent. The age of the patients varies from 18 to almost 80 years. Thus, DHEA-S concentration of each patient should be considered in its reference range (e.g. lower or upper tertile of the reference range)
- The clinical implications are really questionable, and the results should be interpreted with more caution, since DST has been demonstrated to be the best test to recognize hypercortisolism. Moreover, an alternative diagnosis of ACTH dependent CS should be considered also in patients with adrenal nodules or hyperplasia.
Author Response

(The authors gave the same response as above.)

Round 2
Reviewer 2 Report
The authors have addressed my previous requests and suggestions